# Communicating Health Information at the End of Life: The Caregivers’ Perspectives

**DOI:** 10.3390/ijerph16142469

**Published:** 2019-07-11

**Authors:** Olivia Ibañez-Masero, Inés María Carmona-Rega, María Dolores Ruiz-Fernández, Rocío Ortiz-Amo, José Cabrera-Troya, Ángela María Ortega-Galán

**Affiliations:** 1Care Unit, University Hospital Complex of Huelva, 21005 Huelva, Spain; 2Granada District, Andalusian Health Service, 18016 Granada, Spain; 3Department of Nursing, Physiotherapy, and Medicine, University of Almería, 04120 Almería, Spain; 4South Seville HMA, Andalusian Health Service, 41014 Seville, Spain; 5Department of Nursing, University of Huelva, 21007 Huelva, Spain

**Keywords:** information, end of life, humanization, health care system, qualitative research

## Abstract

Health information and communication are key elements that allow patients and family members to make decisions about end-of-life care and guarantee a death with dignity. *Objective*: To understand caregivers’ experiences regarding health information and communication during the illness and death of family members. *Methods*: This qualitative study was conducted in Andalusia based on the paradigm of hermeneutic phenomenology. Participants were caregivers who had accompanied a family member at the end of life for over 2 months and less than 2 years. Five nominal groups and five discussion groups were established, and 41 in-depth interviews with 123 participants were conducted. Atlas.ti 7.0 software was used to analyze the discourses. A comprehensive reading was carried out along with a second reading. The most relevant units of meaning were identified, and the categories were extracted. The categories were then grouped in dimensions and, finally, the contents of each dimension were interpreted and described given the appropriate clarifications. *Results*: Four dimensions of the dying process emerged: differences in caregivers’ perceptions of information and communication, a conspiracy of silence, consequences of the absence or presence of information, and the need for a culture change. *Conclusions*: Poor management of health information and communication at the end of life increased the suffering and discomfort of patients and their families. The culture of denying and avoiding death is still present today. A change in education about death would better enable health professionals to care for patients at the end of life.

## 1. Introduction

Death is a natural process within the life cycle [1]. End-of-life care implies the humane and respectful care of patients and their close family members [2]. Studies focused on death and dying and how health care system providers should facilitate a dignified death are necessary [3]. At the end of life, empathy, tactfulness, showing affection [4,5], and particularly communication between patients, family members, and health professionals are highly valued [6].

Health information and communication about the patient’s prognosis, condition, and treatments administered are key aspects affecting the quality of care received by dying patients [7]. Adequate and honest information allows patients and family members to participate in decision-making processes for necessary end-of-life care [8]. Additionally, the uncertainty about death experienced by patients and their families decreases when fears are able to be expressed to a formal health care provider [9].

The rights of people to receive health information and participate in decision-making processes at the end of life are widely recognized by different international organizations [10]. In Spain, this right is protected by Law 41/2002 on the Autonomy of the Patient and the Rights and Obligations with regard to Clinical Information and Documentation. In Andalusia, these rights are established in the Law on Rights and Guarantees of the Dignity of Persons in the Process of Death, specifically in Title II.

Despite these legal regulations, health care providers continue to avoid providing information to family members and patients about the dying process that would help in decision-making [11]. Sometimes, a “conspiracy of silence” or “pact of silence” occurs in which family members and/or caregivers and professionals decide, not always openly, to withhold information from the patient [12].

Health professionals doubt the advisability of providing accurate information to patients based on the fear of harming patients and the potential legal insecurity [13]. Health care providers feel they lack specific training or skills to manage end-of-life situations, which in turn impacts the quality of care received by patients [14]. On the other hand, patients and family members feel that they are not informed and cannot participate in the decisions that are made regarding necessary care [11]. If they are informed, the attitude is often different―the patients deny or avoid the information, and relatives may maintain an attitude of protection towards the patient so that he or she does not suffer [15].

The consequences of inadequate communication and information are often negative and result in a feeling of patient isolation, some distress in family members, or dissatisfaction with the care received [6]. Very few studies have delved into the experiences of family members with regard to information in the end-of-life process [16]. Therefore, because of the importance of communication and information at the end of life and the resulting consequences, the objective of the present study was to explore caregivers’ experiences regarding health information and communication during the illness and death of their family members.

## 2. Methods

A qualitative research study was performed with a hermeneutic phenomenological approach, according to the Van Manen method [17]. This method allows researchers to study the non-conceptual experiences of people and their meanings. The study was conducted in the region of Andalusia (Spain) from January 2013 to December 2016. A triangulation of qualitative techniques was performed, resulting in 5 nominal groups (*n* = 42) (NGs), 5 discussion groups (*n* = 40) (DGs), and 41 in-depth interviews (IDIs).

Intentional sampling was used [18] in different health care centers (hospitals and primary care). The objective was to select participants who could contribute different perspectives according to the death of their family member, until saturation of the data was reached. The participants were caregivers who had closely accompanied a family member at the end of life. Caregivers of family members who had died in various care settings (home care, hospitalization units, emergency services, intensive care, and palliative care) were included. The death of the relative might have been caused by any condition, including an advanced chronic disease, oncological process, or unexpected death, as long as the patient had been treated by public health professionals. Additionally, death should have occurred between the last 2 months and 2 years to avoid the first stages of grief, and memory of the experience should not have faded over time. All caregivers who were experiencing pathologic grief that could influence the information provided in the interviews were excluded from the study. 

When selecting participants, medical records were consulted after contacting nursing professionals from different health centers, who were informed about the objectives and selection criteria of the study. Subsequently, these professionals interviewed the participants to confirm that they met the selection criteria. Once selected, individuals were invited to participate in the study and were referred to the research team. Feedback between the research team and the professionals who selected the participants continued throughout the study.

First, NGs were developed, followed by DGs, and finally IDIs were conducted [14]. Nominal grouping is a consensus technique allowing researchers to generate hypotheses and obtain criteria by establishing priorities for a need or problem [19]. Therefore, the analysis resulting from the NGs yielded the relevant topics used to prepare the questions to be discussed in the DGs. The aim of this technique was to create a situation of group communication with an enriching discourse. The analysis of the DGs allowed us to develop the questions for IDIs (Table 1). NGs, DGs, and IDIs were conducted by research team members who were previously trained and had no prior contact with study participants as a care provider.

NGs and DGs were conducted in prepared rooms in health centers, with an approximate duration of 60–90 min. Two researchers participated—one stimulating the group and the other recording the observations and incidents in a field notebook. IDIs were conducted in nursing offices or the participant’s home; in the latter case, a researcher from the study visited the participants’ homes to conduct the interviews. The sessions lasted between 45 and 60 min.

The discourses occurring with the different techniques (NGs, DGs, and IDIs) were recorded in audio format and transcribed. Data were independently analyzed by three researchers according to the Giorgi method [20]. First, the texts were comprehensively read. After a second reading, the most relevant units of meaning were identified, and categories were extracted. Then, these categories of meaning or subdimensions were categorized into more general dimensions to form groups. Finally, the contents of each dimension analyzed were interpreted and described. The researchers triangulated the results. Atlas.ti 7.0 software (Scientific Software Development GmbH, Berlin, German) was used to analyze the discourses. The analysis was carried out in Spanish, which is the original language of the caregivers and researchers, and translation into English was performed once the article was written for publication. Approval was obtained from the research ethics committees of the autonomous community of Andalusia in those provinces where the different research techniques were performed. In addition, informed consent was obtained from all participants, and confidentiality and anonymity were maintained throughout the study. At all times, the bioethical principles of the Declaration of Helsinki were respected. Discourse data have been safeguarded, complying with data protection regulations (Organic Law 15/1999, of December 13, on the Protection of Personal Data).

## 3. Results

One hundred and twenty-three caregivers participated, with a mean age of 54.61 years (SD = 10.59 years) and an average duration of care of 9.6 months (SD = 6.6 months). Women represented 86.9% (*n* = 107) of caregivers compared to 13.1% in men (*n* = 16). Table 2 presents the sociodemographic characteristics of the participants.

From the discourse analysis, four dimensions and 10 subdimensions related to health information during illness and death at the end of life emerged (Table 3).

### 3.1. Differences in Caregivers’ Perceptions of Information and Communication

Two very different discourses were observed for caregivers’ perceptions of the health information provided to patients and themselves.

#### 3.1.1. Good Information for the Patient and Family

Overall, caregivers were satisfied with the information provided. They perceived that the information delivered to patients was sufficient, clear, and appropriate to the moment and their needs. Some elements emerging from the discourses and suggesting that positive information was received by the patient and caregivers include respect for the patient’s decision, agreement of all professionals on the information to be provided, its clarity and simplicity, and finally “being tactful”.

“Good. Also, at all times. He was the first one who wanted to know what he had, and he did not back down. The doctors have been frank. The surgeons and all those who attended him”. (IDI P2).

“A: Do you think he was sufficiently or adequately informed? 

  B: Yes 

  A: According to his wishes? 

  B: Yes, yes, yes. It is also that he asked it and they did not deny it”. (IDI P1).

#### 3.1.2. Pace in the Information Provided to Caregivers

In contrast, caregivers had negative perceptions of the amount of information and the rate at which it was provided. Caregivers indicated that the information was provided quickly, without respecting the time caregivers needed to process it. Caregivers argued that they were unable to process so much information of such magnitude and in such a short time. Therefore, although the information was provided, it did not actually or effectively reach the patient or caregivers.

“It’s a lot of information in a very short time. They tell you the news without softening the blow, without explaining things as they should; their information is excessive or falls short. Many times, they talk more than necessary, and you do not know what to do with that information; you are not aware of what they are doing.” (IDI P16).

“He asked, but I think he did not understand what was happening. It’s a lot of information and so sudden.” (IDI P16).

### 3.2. Conspiracy of Silence

Some situations reflect what has been called a “conspiracy of silence” or “pact of silence”. An agreement, sometimes not even explicit, exists between family members and professionals to withhold information from the patient.

“A: Do you think he felt adequately informed of things?

  B: No, because I didn’t want him to be.” (IDI P8).

“And he says: “Well, J., do you know what he has?” And then, I told him: “Sure, he knows what he has...” and then I said: “... but only half of it”. My husband trusted me. I do not know if I misled him very well, or he had full faith in me, he trusted everything I told him.” (IDI P6).

#### 3.2.1. Caregivers’ Reasons for Withholding Information

Occasionally, caregivers do not inform relatives about the disease process they are experiencing. This lack of information is generally motivated by a protective desire of caregivers. They hold the belief patients will suffer more if the truth is known. They hid the information, looking for the best situation for everyone and assuming it was what all wanted, or at least, what caregivers wanted. In general, caregivers think that what they believe to be the best for themselves is always the best for their loved ones.

“I used to say: “Look, P., this pill is for that, and this one here is for splitting. This one is for going to the bathroom, and this one for...” And he, he took them with such faith... I believe I acted as he wanted me to.” (IDI P6).

“P. knew what he had, I had it wrapped in a coloured paper. Do you understand me? The biggest hurdle for him was when he had to stop driving, and I told him: “P., this situation will not last forever, things do not go on forever.” He never thought it was his end... Because no, no. No. It is the same I want for me. I do not want to know that much.” (IDI P6).

#### 3.2.2. Patients’ Reasons for Silence

Regarding patients, the discourses reflected that they sometimes chose silence and did not communicate with other family members. The individuals involved occasionally avoid speaking calmly and clearly about the disease processes in the immediate, intimate and family environment. Patients try to protect their families because they do not wish to cause additional suffering to the people closest to them.

“He never asked, no. He behaved just like my father, who also did not ask although he knew the situation. He asked my mother, but not us. It seemed as if he wanted to take away the problem from us.” (IDI P16).

“With us, he wanted to talk about our life, about us, about... and about the future.” (IDI P12).

“She was a very private person and kept things to herself; she would not tell you what she had, so others would not suffer.” (DG P4).

#### 3.2.3. Evaluation of Silence in Professionals

Caregivers excuse withholding information or being silent at the end of life. They think the explanation for this silence, favored by practitioners, helps to avoid the unnecessary suffering of the patient and their families and to maintain the hope and expectation of recovery.

“For six months I lived with the illusion we would overcome the situation, and if they had told me the prognosis was poor, my attitude would not have been the same. When health personnel entered the room, they said there was light and joy. I thought we would be able to leave here.” (IDI P16).

“They might have hidden things from us so we would not suffer.” (IDI P19).

### 3.3. Consequences of the Absence or Presence of Information

#### 3.3.1. Patient Isolation

The conspiracy of silence leads to situations in which the patient and family members suffer throughout the process in solitude. The patient can remain isolated because the family does not agree on the best way to manage information. Caregivers perceive that patients knew what is happening, but they cannot communicate with each other. For the relatives, this context resulted in the patient dying unaware of what was happening, ceasing to participate at the end of life, and hindering communication and the last farewell of loved ones. The patient remained isolated because the family did not agree on the best way to manage information. Caregivers reported that patients felt they were being deceived about the situation, and therefore they demanded and almost begged for clear information about their condition.

“She said, “I’m going to die, right?” Because nobody told her she was going to die, and I thought she should be told, and at last she asked me, and I could not say no. Really, was I going to make up another story? That’s why I stress the importance of the family agreeing on communication, because if not....” (NG P3).

#### 3.3.2. Complicated Grief

Silence and the concept of death as a taboo topic extend to the grieving process. Caregivers mentioned the difficulty of speaking openly about the subject with family members. Unresolved grieving occurred, resulting in negative experiences.

“I do not know if it would be good, but in my home, it was taboo (...) My little brother passed away and, as time went by, nobody said anything, nothing was talked about.” (IDI P3).

The conspiracy of silence also caused uncertainty because of a lack of first-hand knowledge about the desires of the dying person. This process favored emotional blockage, guilt, and the subsequent development of complicated grief.

“Many times, I think: “have I left something pending or unresolved?” And I cannot stop thinking about it over and over in my head.” (NG P4).

#### 3.3.3. Benefits of Open Communication

However, when professional information, communication, and guidance broke the conspiracy of silence, caregivers and patients at the end of life reported great satisfaction. It is a frank, sincere and adequate communication that allowed both of them to feel free to express themselves and satisfy the final needs and desires of the dying person. Open communication was a key element in the experience. The process changed from one generating great distress to a pedagogical, vital, and unique process, giving new meaning to pain.

“At first I didn’t want him to know anything. She asked and I kept quiet, but then I thought it shouldn’t be like. She received the help of a nurse experienced in this area, who was a great professional and could guide me.” Then, my friend and I experienced a change.... She wanted to die in my company because she had told me so; without sedating her, or any other actions, she died accompanied by me, she passed away quietly, and it turned out well. It could have gone wrong, but it went well. So, for me, it was something magical... I was relieved. It was an experience... sad, but... my satisfaction is beyond words.” (NG P2).

### 3.4. Need for a Paradigm Change in the Dying Process

Caregivers perceived that death is still considered a topic that is forbidden to discuss or ponder [21]. The open discourse about the approaches, doubts, and fears surrounding death as a human experience is not encouraged. Informants believed that open discourse about the approach, doubts, and fears surrounding death as a human experience is not encouraged. This cultural imprint has also entered the health system and influences professional training. The participants felt that the ability to learn to accept and accompany death as part of life is often focused on fighting against it.

#### 3.4.1. Preparing for Death

Participants expressed the need for society to include death and loss in the learning process. As a result, people would be prepared to experience these processes in a more natural and adaptive way―with less deception and silence―by openly communicating. From the perspective of caregivers, the incorporation of these changes would improve the farewell to the dying person.

“[Resolving] all the shortcomings of a system not preparing us for death, that would relieve us a lot, both for the departing loved one, and for those of us helping them to leave, in a much happier way.” (NG P5).

#### 3.4.2. Need for Professional Training

Additionally, the discourses also included the need to train professionals to communicate information during the dying process and manage the care provided in situations of death and subsequent grief.

“Training is very important; training to communicate information, I think it’s fundamental. The information must be given by professionals to the patient. And then to the family. Professionals must know the whole family suffers when there is a terminal illness, and the situation generates a conflict (...) This is not managed by the health services.” (NG P5).

## 4. Discussion

Regarding end-of-life patient care, caregivers emphasize the importance of communication and the provision of information between the patient, family members, and professionals involved, as reported in different studies [22,23]. Deficiencies in communicating health information at the end of life exist. The conspiracy of silence is a dynamic established at the end of life that is motivated by concern for the patient and the desire to protect them from further suffering.

Although patients at the end of life have the legal right to information about their situation, the difficulty to transmit this information has been reiterated by caregivers [5,15,24]. Adequate information facilitates the decision-making process and reduces suffering by reducing uncertainty and enabling compliance with the wishes of the person at the end of life [9,25].

In the initial phases of adaptation and coping, the patient and caregiver may develop an attitude called a “conspiracy or pact of silence”, avoiding talking or inquiring about the condition; this attitude seems to be very normal and repetitive [6,16]. Caregivers feel the need to protect the patient’s emotions and might not talk about related issues or the real concerns of patients, strengthening the conspiracy of silence [26]. Moreover, professionals tend to adjust to this process. Therefore, their abilities to engage in authentic and serene communication with the patient and family members are hampered because they do not actually know the true needs or desires of patients at the end of life [6].

Consistent with the findings reported by Epstein et al. [10], the lack of communication and information isolates patients at the end of life, hindering them from closing pending issues, which increases their suffering. The ability to say goodbye emerges as a key element favoring peace, both for patients in their last moments and for the companions in preparing for subsequent grieving [11]. This process is intimately related to clear and authentic communication, breaking the pacts of silence [6,27].

Patients and their families demand greater professional skill in communicating death-related bad news and a more humanized treatment, attributes that are directly related to the quality of care provided during the dying process [21]. The most common position in the accompaniment of this process oscillates between the abandonment of the professional who abruptly delivers the information, without exploring the wishes of the patient and family members, and the pact with the family and friends to withhold information about the current diagnosis and prognosis, making it difficult for the moribund to adapt to the process [28]. However, open communication spaces that favor the relationship between patients and their families must be created to promote a framework of accompaniment based on truth and respect for autonomy [29,30].

Notably, the difficulties for this accompaniment to the patient and family are not only related to professional training but also to the culture of death denial [21]. Compliance with state and regional laws is the minimum ethical requirement. However, after several years of providing care in the context of a regional regulation prohibiting the concealment of information from the patient, professionals continue to collaborate with the pacts of silence established by families, and the families see this collaboration as positive [26]. Duty ethics with normative development do not appear to be sufficient for developing the legal right. Work at the social level is needed to change the misconceptions and beliefs surrounding death [31]. Professionals should reflect on the need to change the cultural patterns in which they are immersed regarding the dying process. They should cultivate the basic attitudes and virtues necessary for the transmission of good information and communication at the end of life [32].

The triangulation of different techniques and researchers has provided methodological rigor to the study, but certain limitations may exist [33]. The present study includes a broad sample of family caregivers, mostly women. Gender inequality, a characteristic of the informal care system, might have influenced caregivers’ experiences [34]. On the other hand, the cultural level of study participants has not been considered. This aspect might impact the satisfaction of the perceived experience. Additionally, the experiences were not analyzed according to the level of care in which the relatives attended. However, the wide variability of subjects receiving care from health professionals in different contexts and situations suggests the great strength of the present study in terms of the results obtained. Another limitation was caregivers with pathologic grief that could influence the information were excluded. Even so, in the discourses appeared a dimension of complicated grief that was not excluded from the analysis and it was a consequence that the lack of information had on the caregivers.

As a future direction of research, an increase in the effectiveness of the methods facilitating health care provider–patient–family communication would be useful, allowing practitioners and caregivers to promote health and maintain the quality of life as much as possible during the dying process. Studies that, in addition to determining the effectiveness of care, allow a reconciliation of the professionalism and humanization of care, along with managing feelings during the dying process, are needed.

## 5. Conclusions

Differences in caregivers’ perceptions of patients at the end of life regarding the communication and information provided by family members were observed. The lack of emotional support caused by the poor communication and information provided by health professionals distresses patients and family members. Caregivers who have been helped by professionals trained in this area feel deeply grateful, acknowledge the good work and the positive repercussions for themselves and the patient.

The conspiracy of silence is an established dynamic at the end of life. This practice is motivated by concern for patients and the desire to protect them from further suffering. However, these beliefs are based on a culture of denying death and a prohibition of any discourse about death. A culture change in the attitudes towards death, which includes the dying process in the collective imagination in a natural and inevitable way, would enable people to prepare themselves for death and would encourage the health professions to be trained for accompaniment at the end of life.

## Figures and Tables

**Table 1 ijerph-16-02469-t001:** Question script: nominal groups (NGs), discussion groups (DGs), and in-depth interviews (IDIs).

NGs	DGs	IDIs
What aspects of the health care system hindered or facilitated the dying process of your relative?	How do you think the people you helped felt at the end of life?	Regarding information about the disease, what do you think about the information the patient received?
	What do you think about the care they received?	How would you rate the method in which the information was provided?
	What aspects would you highlight regarding the information provided by health professionals?	How did the patient feel about the information he/she received?
		How did you feel?
		How did the information received from health professionals help the patient make decisions?
		How do you think the information received affected your relative at the end of life?
		How did the information affect you?

NGs = Nominal groups; DGs = discussion groups; IDIs = in-depth interviews.

**Table 2 ijerph-16-02469-t002:** Sociodemographic characteristics of the participants (%).

Variables	NGs (*n* = 42)	DGs (*n* = 40)	IDIs (*n* = 41)
Gender			
Women	94.6	78.5	87.8
Men	5.4	21.5	12.2
Age (years)			
>49	35.4	40	34.1
50–69	54.6	52.5	61
>70	10	7.5	4.9
Kinship			
Son/daughter	29.7	53.3	4.9
Spouse	35.2	26.6	29.3
Father/mother	21.6	13.3	46.3
Brother/sister	8.1	6.6	9.8
Others	5.4	0.2	9.7
Place of death			
Hospital	51.3	48.2	56.1
Home	40.5	41.3	41.5
Street	5.4	6.9	2.4
Others	2.8	3.6	0

NGs = nominal groups; DGs = discussion groups; IDIs = in-depth interviews.

**Table 3 ijerph-16-02469-t003:** Dimensions and subdimensions emerging from the study.

Dimensions	Subdimensions
Differences in caregivers’ perceptions of information and communication	Good information was provided to the patient and family.
	Pace in the information provided to caregivers.
Conspiracy of silence	Caregivers’ reasons for not wanting to provide information.
	Patients’ reasons for silence.
	Evaluation of silence in professionals.
Consequences of the absence or presence of information	Patient isolation.
	Complicated grief.
	Benefits of open communication.
Need for a paradigm change regarding the end of life	Preparing for death.
	Need for professional training.

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
