# Peer review of "Communicating Health Information at the End of Life: The Caregivers’ Perspectives"

_ijerph, 2019, doi:10.3390/ijerph16142469_

Reviewer 1 Report

Congratulations, I personally find this study very well conducted; in particular, the Methods are well selected and the process of textual Analysis thoroughly presented. Therefore, the results are meaningful, indeed.

Hence, I have few minor comments:

1.     Please modify the abstract, so that presents also the textual Analysis part of your study.

2.     Please add a comment on, in which language the textual Analysis was conducted and at what stage were the examples translated into English

3.     What you call a conspiracy; Foucault has defined as meaningful silence (There is not one but many silences, and they are an integral part of the strategies that underlie and permeate discourses.). As you use phenomenology and speak about discourse, perhaps the concept of meaningful silence could be considered? or is the conspiracy a different thing?

Author Response

Reviewer 1

1. Please modify the abstract, so that presents also the textual Analysis part of your study.

We have added the following paragraph to the abstract:

A comprehensive reading was carried out along with a second reading. The most relevant units of meaning were identified, and the categories were extracted. The categories were then grouped in dimensions, and finally, the contents of each dimension were interpreted and described given the appropriate clarifications.

2. Please add a comment on, in which language the textual Analysis was conducted and at what stage were the examples translated into English

We have added in the methods the following paragraph:

The analysis was carried out in Spanish, which is the original language of the caregivers and researchers, and translation into English was performed once the article was written for publication.

3. What you call a conspiracy; Foucault has defined as meaningful silence (There is not one but many silences, and they are an integral part of the strategies that underlie and permeate discourses.). As you use phenomenology and speak about discourse, perhaps the concept of meaningful silence could be considered? or is the conspiracy a different thing?

The concept of “Conspiracy of Silence” does not refer to silence itself but rather to the fact that the patient does not receive health information regarding the disease process. There are very few true silences in conspiracy-driven relationships, because these silences are full of empty words or lies. A parallel story is seen in the disease process that even the relatives and the patient him or herself believes; this would not be questioned if not for the fact that physical deterioration is unrelenting. Therefore, it is different from Foucault’s concept, which refers more to actual silence and its meaning.

Reviewer 2 Report

It is not clear to me how many carers did not want information to be given to the dying person, The conclusions seem to suggest that education would be the solution. If carters as well as professionals believe that telling a dying person the information will be harmful to them how will education help.

Author Response

Reviewer 2

It is not clear to me how many carers did not want information to be given to the dying person. The conclusions seem to suggest that education would be the solution. If carters as well as professionals believe that telling a dying person the information will be harmful to them how will education help.

 Regarding the second issue, it should be clarified that the right to health information at the end of life is regulated by law, so that professionals are obliged to know and comply with it and require training in this regard. If we start from a deeper socio-cultural analysis, the tendency not to give information to professionals and caregivers is rooted in the paternalistic paradigm and in the social way of facing death as a taboo. Both elements involve professionals and caregivers, as they are included in this cultural pattern. The educational proposal is in the social key of the normalization of the process of death and the recognition of personal autonomy for health decision making. This transformation requires including these new norms in the educational process from infancy and in the training of future professionals.

Reviewer 3 Report

Communicating health information at the end of life:  the carer’s perspective

The authors are to be commended for working in an area that is so important – carer’s working with seriously ill patients.  The effort placed to capture themes and distill them is to be applauded.  I hope my comments help to strengthen this paper and eventually allow it to be suitable for publication. 

Abstract:

Line 15 – ‘… about end-of-life process…’ is awkward phrasing.  Maybe substitute ‘care’ for ‘process’.

Line 25 – ‘ … at the end of life suffering… ‘ wondering if you are missing ‘increased’ between ‘end of life’ and ‘suffering’

Introduction:

Line 33 – define ‘formal care setting’

Second paragraph does not appear to be on point for the idea of this manuscript.  Rest of the intro covers more relevant topics in review. 

Overall, the intro does not help the reader know how this manuscript adds to the literature.  As written, it appears a lot is known about this.  For example, if a lot was known about health info and communication from clinician and pt perspective but not carer’s perspective, then highlight this in the intro. 

Methods:

Line 71 – describe more about what study team did for intentional sampling – which clinics, hospitals did they reach out to and why.  What where they trying to intentionally sample? 

Line 78 – how did the study team determine ‘complicated grief’ to exclude? 

Lines 78-81 – this text and Table 1 belong in Results and not Methods. 

Table 1 – DGs, NGs, and IDIs need to be spelled out in table so a reader could understand the table without reading the text.  Same feedback for Table 2.

Line 105 – was this process only for IDI interviews or also NG and DG interviews?

Was thematic saturation reached?  This helps the readers know how close the study team may have gotten to a ‘full picture’ from their interviews. 

Results:

Table 3 – do subdimensions fit under dimensions?  It is unclear from the table if these are connected concepts or all separate.  Also, ‘inadequate information’ in the text (line 136) seems to indicate more about the tenor and style which the info was communicated and less evidence for a lack of info from professionals as written. 

Line 134 – this quote does not seem to represent the dimension very well

Line 137 – can the authors give context for what ‘some’ means?  Given carer’s were happy with info it would be good to know how many were unhappy so readers can understand the potential magnitude of this issue. 

Line 152 – this quote needs to be tightened up, too long and rambly as written and hard to get the main point authors are trying to represent.

Line 187 – this section is confusing.  It is written as if it was established from interviews with professionals as it speaks to their motivations, so the authors need to be VERY clear that these are the hypotheses of the carer’s – and even then this reviewer is not sure this belongs in this manuscript.  If the study team wanted the professional’s perspective, they can conduct interviews. 

Line 204 – how did the carer’s know that the patients felt they were being deceived?  Authors at least need to indicate clearly that the carer’s felt the patients felt they were being deceived.  Written in this way, it does appear like patients should be interviewed if this perspective is something this research team wanted to understand. 

Line 210 – complicated grief is a dimension but the authors indicate they excluded carer’s with complicated grief.  Help me understand the rationale here?  I worry this is a sign they didn’t exclude the population they intended to. 

Line 223 – more info = greater satisfaction is not what the authors report earlier in the manuscript (around line 136) so please clarify this statement/perspective.

Line 227 – this quote is not representative of professional information delivered which is what the paragraph before highlights.  Find a better quote to represent this point. 

Line 233 – this is written from the voice of the authors and should be written representing what was said during interviews by carers. 

Author Response

Reviewer 3

Abstract:

Line 15 – ‘… about end-of-life process…’ is awkward phrasing. Maybe substitute ‘care’ for ‘process’.

We have added “care”.

Line 25 – ‘ … at the end of life suffering… ‘ wondering if you are missing ‘increased’ between ‘end of life’ and ‘suffering’

We have added “increased”.

Introduction:

Line 33 – define ‘formal care setting’

Second paragraph does not appear to be on point for the idea of this manuscript. Rest of the intro covers more relevant topics in review.

Overall, the intro does not help the reader know how this manuscript adds to the literature. As written, it appears a lot is known about this. For example, if a lot was known about health info and communication from clinician and pt perspective but not carer’s perspective, then highlight this in the intro.

The suggested paragraph has been deleted and other elements that could clarify the introduction have been introduced.

Methods:

Line 71 – describe more about what study team did for intentional sampling – which clinics, hospitals did they reach out to and why. What where they trying to intentionally sample?

We have added the following paragraph:

Intentional sampling was used18 in different health care centers (hospitals and primary care). The objective was to select participants who could contribute different perspectives according to the death of their family member, until saturation of the data was reached.

Line 78 – how did the study team determine ‘complicated grief’ to exclude?

We have added the following paragraph:

All caregivers who were experiencing pathologic grief that could influence the information provided in the interviews were excluded from the study.

Lines 78-81 – this text and Table 1 belong in Results and not Methods.

We have moved this text and Table 1 to the Results section.

Table 1 – DGs, NGs, and IDIs need to be spelled out in table so a reader could understand the table without reading the text. Same feedback for Table 2.

We have added the appropriate clarifications for Table 1 and Table 2.

Line 105 – was this process only for IDI interviews or also NG and DG interviews?

Was thematic saturation reached? This helps the readers know how close the study team may have gotten to a ‘full picture’ from their interviews.

We have added information.

The discourses occurring with the different techniques (NGs, DGs, and IDIs) were recorded in audio format and transcribed.

The objective was to select participants who could contribute different perspectives regarding the death of their family member, until the saturation of the data was reached.

Results:

Table 3 – do subdimensions fit under dimensions? It is unclear from the table if these are connected concepts or all separate. Also, ‘inadequate information’ in the text (line 136) seems to indicate more about the tenor and style which the info was communicated and less evidence for a lack of info from professionals as written.

The table has been modified. We have changed the subdimension “Inadequate information” for. “Speed in the information provided to caregivers”. We believe this term is more appropriate.

Line 134 – this quote does not seem to represent the dimension very well.

A different quote has been used.

“A: Do you think he was sufficiently or adequately informed?

B: Yes

A: According to his wishes?

B: Yes, yes, yes. It is also that he asked it and they did not deny it". (IDI P1).

Line 137 – can the authors give context for what ‘some’ means? Given carer’s were happy with info it would be good to know how many were unhappy so readers can understand the potential magnitude of this issue.

We cannot quantify the number of caregivers who were satisfied or dissatisfied with the information, since in qualitative research we work with discourses. In the analysis we found statements with both perspectives.

“Some” has been withdrawn because it may cause confusion.

Line 152 – this quote needs to be tightened up, too long and rambly as written and hard to get the main point authors are trying to represent.

The quote has been changed to this one.

“A: Do you think he felt adequately informed of things?

B: Not, because I didn't want to.” (IDI P8).

Line 187 – this section is confusing. It is written as if it was established from interviews with professionals as it speaks to their motivations, so the authors need to be VERY clear that these are the hypotheses of the carer’s – and even then this reviewer is not sure this belongs in this manuscript. If the study team wanted the professional’s perspective, they can conduct interviews.

The text has been modified; we think it was a translation error since the interviews were conducted with caregivers, not professionals.

Cares excuse withholding information or being silent at the end of life of the health care professionals. They think the explanation for this silence favoured by practitioners is to avoid the unnecessary suffering of the patient and their families and to maintain the hope and expectation of recovery.

Line 204 – how did the carer’s know that the patients felt they were being deceived? Authors at least need to indicate clearly that the carer’s felt the patients felt they were being deceived. Written in this way, it does appear like patients should be interviewed if this perspective is something this research team wanted to understand.

“Conspiracy of silence” or “pact of silence” occurs when family members and/or carers and professionals decide, not always openly, to hide information from the patient.

We have modified the text to clarify this concept.

The conspiracy of silence leads to situations in which the patient and family members suffer throughout the process in solitude. The patient remained isolated because the family did not agree on the best way to manage information. Cares perceive patients knew what was happening but could not communicate with each other. For the relatives, this context resulted in the patient dying unaware of what was happening, ceasing to participate at the end of life, and hindering communication and the last farewell of loved ones. Cares report patients felt they were being deceived about the situation, and therefore they demand and almost beg for clear information about their condition.

Line 210 – complicated grief is a dimension but the authors indicate they excluded carer’s with complicated grief. Help me understand the rationale here?. I worry this is a sign they didn’t exclude the population they intended to.

We have added information to clarify which population was excluded from the study.

All caregivers who were experiencing pathologic grief that could influence the information provided in the interviews were excluded from the study.

Line 223 – more info = greater satisfaction is not what the authors report earlier in the manuscript (around line 136) so please clarify this statement/perspective.

In this dimension we are referring to the communication provided by health professionals that breaks the conspiracy of silence process. It is an open and sincere communication that produces satisfaction in the caregivers. We have modified the text to clarify this dimension.

However, when professional information, communication, and guidance broke the conspiracy of silence, caregivers and patients at the end of life reported great satisfaction. It is a frank, sincere and adequate communication that allowed both of them to feel free to express themselves and satisfy the final needs and desires of the dying person.

Line 227 – this quote is not representative of professional information delivered which is what the paragraph before highlights. Find a better quote to represent this point.

A part of the speech that could have been confusing has been eliminated and some lines have been included in the speech that make it more understandable:

“At first I didn't want him to know anything. She asked and I kept quiet, but then I thought it shouldn’t be like. She received the help of a nurse experienced in this area, who was a great professional and could guide me.” Then, my friend and I experienced a change.... She wanted to die in my company because she had told me so; without sedating her, or any other actions, she died accompanied by me, she passed away quietly, and it turned out well. It could have gone wrong, but it went well. So, for me, it was something magical... I was relieved. It was an experience... sad, but... my satisfaction is beyond words.” (NG P2).

Line 233 – this is written from the voice of the authors and should be written representing what was said during interviews by carers.

We have modified the text.

Caregivers perceived that death is still considered a topic that is forbidden to discuss or ponder21. The open discourse about the approaches, doubts, and fears surrounding death as a human experience is not encouraged. Informants believed that open discourse about the approach, doubts, and fears surrounding death as a human experience is not encouraged. This cultural imprint has also entered the health system and influences professional training. The participants felt that the ability to learn to accept and accompany death as part of life is often focused on fighting against it.

Round  2

Reviewer 3 Report

This manuscript has been significantly improved and the authors are to be commended.  

My only available minor recommendations:

Speed of info may be better labeled as 'Pace' of info.  

This reviewer continues to be confused by the exclusion of caregivers with complicated grief yet a theme found was complicated grief.   Please be more explicit with what tool or criteria was used to exclude caregivers with complicated grief.  If authors tried to exclude, but they still found complicated grief as a theme, then this should be a limitation.  That said, this reviewer is grateful complicated grief caregivers were not excluded since it is this population that is of interest to prevent.  

Author Response

Speed of info may be better labeled as 'Pace' of info.

We have added “Pace”.

This reviewer continues to be confused by the exclusion of caregivers with complicated grief yet a theme found was complicated grief. Please be more explicit with what tool or criteria was used to exclude caregivers with complicated grief. If authors tried to exclude, but they still found complicated grief as a theme, then this should be a limitation. That said, this reviewer is grateful complicated grief caregivers were not excluded since it is this population that is of interest to prevent.

We have added the following paragraph:

Another limitation was caregivers with pathologic grief that could influence the information were excluded. Even so, in the discourses appeared a dimension of complicated grief that was not excluded from the analysis and it was a consequence that the lack of information had on the caregivers.